# Leveraging deep active learning to annotate the first public dataset for identification of mobility functioning information in clinical text

Tuan-Dung Le
*Department of Computer Science and Engineering*
*University of South Florida*
Tampa, FL
tuandungle@usf.edu

Zhuqi Miao
*Department of Management Science and Information Systems*
*Center for Health Systems Innovation*
*Oklahoma State University*
Stillwater, OK
zhuqi.miao@okstate.edu

Samuel Alvarado
*Center for Health Sciences Innovation*
*Oklahoma State University*
Stillwater, OK
sam.rubrum@gmail.com

Brittany Smith
*Center for Health Sciences Innovation*
*Oklahoma State University*
Stillwater, OK
brittani.smith10@okstate.edu

William Paiva
*Spears School of Business*
*Center for Health System Innovation*
*Oklahoma State Universiy*
Stillwater, OK
wpaiva@okstate.edu

Thanh Thieu
*Department of Machine Learning*
*Moffitt Cancer Center*
Tampa, FL
thanh.thieu@moffitt.org

*Abstract*—Function is increasingly recognized as an important indicator of whole-person health, although it receives little attention in clinical natural language processing research. We introduce the first public annotated dataset specifically on the Mobility domain of the International Classification of Functioning, Disability and Health (ICF), aiming to facilitate automatic extraction and analysis of functioning information from free-text clinical notes. We utilize the National NLP Clinical Challenges (n2c2) research dataset to construct a pool of candidate sentences using keyword expansion. Our active learning approach, using query-by-committee sampling weighted by density representativeness, selects informative sentences for human annotation. We train BERT and CRF models, and use predictions from these models to guide the selection of new sentences for subsequent annotation iterations. Our final dataset consists of 4,265 sentences with a total of 11,784 entities. The inter-annotator agreement (IAA), averaged over all entity types, is 0.72 for exact matching and 0.91 for partial matching. We train and evaluate common BERT models and state-of-the-art Nested NER models. The best F1 scores are 0.83 for Action, 0.69 for Mobility, 0.60 for Assistance, and 0.67 for Quantification. Empirical results demonstrate promising potential of NER models to accurately extract mobility functioning information from clinical text. The public availability of our annotated dataset will facilitate further research to comprehensively capture functioning information in electronic health records (EHRs).

*Index Terms*—functional status information, mobility, clinical notes, n2c2 research datasets, natural language processing

This study is funded by grant #HR21-173 through the Oklahoma Center for the Advancement of Science and Technology.

## I. INTRODUCTION

Functional status refers to the level of activities an individual performs in their environment to meet basic needs and fulfill expected roles in daily life [1]. It is increasingly recognized as an important health indicator in addition to mortality and morbidity [2], [3]. Since function is not well perceived in medical coding, most functioning information is hidden in free-text clinical notes. However, Natural Language Processing (NLP) research on the secondary use of EHRs has focused primarily on health conditions (ie, diseases, disorders) and related drugs [4]. Automatically extracting and coding functioning information from clinical text is still a relatively new and developing field in the NLP community, and there is a critical need to develop resources and methods to advance research in this area.

Function is a broad ontology defined by the International Classification of Functioning, Disability, and Health (ICF) [5] - a classification system developed by the World Health Organization (WHO) with the aim of standardizing the description of health and health-related states. Even though ICF is less applied in practice than its sister International Classification of Diseases (ICD), we utilize this framework to code mobility function because it is the only functional taxonomy backed by the WHO. Previous studies [6]–[10] have focused mainly on the Mobility domain of the ICF due to its well-defined and

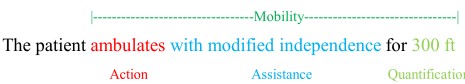

The patient ambulates with modified independence for 300 ft

Action     Assistance     Quantification

Fig. 1. An example with nested annotation for Mobility, Action, Assistance and Quantification.

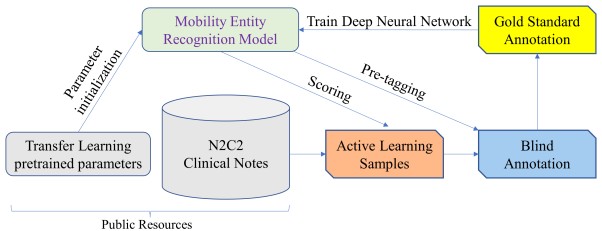

Fig. 2. Our iterative deep active-transfer learning

observable nature as a construct of human functioning. Thieu et al [6], [10] constructed a private dataset from 1,554 physical therapy (PT) notes provided by the National Institutes of Health (NIH) Biomedical Translational Research Information System and deduced a fine-grained hierarchy between nested mobility-related entities (Fig. 1): Mobility is a self-contained description of physical functional status, Action captures the activity, Assistance includes information about supporting devices or persons, Quantification details measurement values, and Score Definition provides standardized assessments, often as numerical values. They developed named-entity recognition (NER) models on this dataset and achieved 84.90% average F1 score. However, there exist limitations: (1) the unavailability of the private corpus hinders research collaboration with the public community; and (2) it is unknown how well the models perform beyond NIH data with different institutional language idiosyncrasies.

To address these limitations, we explore the publicly available National NLP Clinical Datasets (n2c2) [11], which is contributed by Partners Healthcare consisting of 15 hospitals and healthcare institutes, ensuring robustness to institutional language idiosyncrasies. Unfortunately, the n2c2 data lacks mobility annotations, making them unsuitable for supervised entity recognition methods. Furthermore, manually annotating mobility-related entities by domain experts is costly at scale. Deep active learning algorithms were designed to mitigate this problem by strategically choosing the examples to annotate, aiming to obtain better downstream models with fewer annotations [12]–[15].

In this work, we employ deep active learning to create a public mobility entity dataset and develop NER models with n2c2 data. We use pool-based [16] query-by-committee sampling [17] weighted by density representativeness [18] to select the most informative sentences for human annotation. Our committee models, BERT [19] and CRF [20], were chosen based on their complementary characteristics: BERT achieves high recall but low precision, while CRF offers high precision but low recall for identifying mobility entities, as demonstrated in a previous NIH study [10]. These differences ensure diverse predictions, which are critical for our proposed active learning strategy.

Our contributions can be summarized as follows:

- We create the first publicly available mobility NER dataset for the research community to extract and analyze mobility information in clinical notes.
- We provide the baseline evaluation results on our dataset using state-of-the-art NER approaches.

## II. RELATED WORKS

### A. Functional status information (FSI)

Due to the lack of a standardized functioning ontology [21] and the incompleteness of the ICF as a vocabulary source [22], previous studies rely on clinical staff to collect function phrases through focus groups [23], [24] or manual chart reviews [21], [25]. Kuang et al [21] manually gathered patient-reported function terms from clinical documents and online forums, facing challenges in matching them with Unified Medical Language System terms. Additionally, there were few attempts to automatically identify FSI from clinical notes. Those methods were limited to specific ICF codes [26] or relied on ad-hoc mapping tables [23] to alleviate the absence of a repository containing function-related concepts. Newman-Griffis et al [27] emphasized the importance of capturing FSI in healthcare systems and called for more research in this area.

### B. Mobility domain within ICF

Thieu et al [6] took the initial step towards extracting FSI from clinical notes by systematically identifying Mobility-related FSI. They first created a dataset of 250 de-identified PT notes, including details about the activity being performed, sources of assistance required, and any measurements described in the notes. Expanding to 400 PT notes [10], they achieved high performance in Mobility NER using an ensemble of CRF, RNN, and BERT models, showing the efficacy of their approach with sufficient resources. Other attempts on this dataset explored domain adaptation of mobility embeddings [7], action polarity classification [8] linking action to ICF codes [9]. However, this dataset is private and thus restricted to only a handful of NIH researchers. Recently, Zirikly et al [28] introduced publicly available dictionaries of terms related to mobility, self-care, and domestic life to facilitate the retrieval and extraction of disability-relevant information. These terms were curated from NIH and Social Security Administration documents, and their performance on other institutional data remains untested.

## III. METHODS

In this section, we present our deep active learning framework (Figure 2) for incrementally developing Mobility NER models together with gold-standard annotated datasets using the n2c2 research dataset. We detail the pre-processing, data retrieval, annotation, and active learning strategy.

## A. Data Collection

*1) Data source selection:* Two most well-known dataset that provide clinical notes for research purposes are MIMIC [29], [30] and the National NLP Clinical Challenges (n2c2) [11]. While the MIMIC dataset has driven large amount of research in clinical informatics, its limited scope - only including data from patients in critical care units at one institution - makes it less suitable for addressing the institutional language idiosyncrasies problem. In contrast, the n2c2 dataset offers greater diversity in linguistic styles with data from 15 hospitals and healthcare institutes. Its 2018 dataset contains 505 discharge summaries from MIMIC-III. Although the MIMIC dataset, particularly the newer MIMIC-IV version, contains a larger number of cases and documents, utilizing such a large dataset would significantly increase the computational cost of scoring all candidate sentences during the active learning process. By selecting n2c2, we strike a balance between computational feasibility and the diversity of institutional language captured.

In addition, n2c2 also originates from the north-east area of the U.S., thus it might share similar scribing style with clinical notes from the NIH, making re-using of the annotation guideline from a previous work based on NIH clinical notes [10] more effective (see Section III-B). In this study, we utilize the n2c2 research datasets, which comprise unstructured clinical notes from the Research Patient Data Registry at Partners Healthcare, originally created for the i2b2 annual shared-task challenge projects from 2006 (Table I). We obtained a total of 6,614 text notes by downloading all available datasets from the DBMI Data Portal[1].

*2) Data Preprocessing:* Our work utilizes n2c2 notes, primarily discharge summaries, where we observe a sparser occurrence of mobility information compared to NIH PT notes used in previous study [6], [10]. To mitigate the labor-intensive process of scanning through a large volume of irrelevant text, we perform sentence-level annotation selected by active learning instead of note-level annotation. Active learning requires re-scoring the entire pool of unlabeled data at each iteration. We employ de-duplication and keyword expansion that filter out mobility-relevant sentences to reduce the size of the unlabeled sentence pool. Specifically, remaining n2c2 sentences after deduplication are indexed into Lucene [31], a high-performance text search engine. We define mobility-relevant keywords by extracting terms from the domain and subdomain definitions (including inclusions) under the "d4 Mobility" section of the ICF framework[2]. Next, we filter out NLTK stop words [32] and irrelevant words, and then expand the set with inflections[3] to create the first keyword set $K = \{k_1, k_2, .., k_n\}$. Retrieved sentences are obtained from Lucence using query: $k_1$ OR $k_2$ OR ... OR $k_n$.

Since short definitions from ICF do not include all possible mobility-relevant keywords, we improve sentence retrieval

---

[1] https://portal.dbmi.hms.harvard.edu/projects/n2c2-nlp/
[2] https://icd.who.int/dev11/l-icf/en
[3] https://pypi.org/project/pyinflect/

---

TABLE I
THE NATIONAL NLP CLINICAL CHALLENGES (N2C2)'S RESEARCH
DATASETS.[1]

| Year | Dataset Name | Size |
|------|--------------|------|
| 2006 | Deidentification & Smoking | 889 discharge summaries |
| 2008 | Obesity | 1237 discharge summaries |
| 2009 | Medication | 1243 discharge summaries |
| 2010 | Relations | 1748 discharge summaries and progress reports [2] |
| 2011 | Coreference | 978 discharge summaries, progress notes, clinical reports, pathology reports, discharge records, radiology reports, surgical pathology reports, and other reports |
| 2012 | Temporal Relations | 310 discharge summaries |
| 2014 | Deidentification & Heart Disease | 1304 longitudinal medical records |
| 2018 | Track 1: Clinical Trial Cohort Selection | 1304 discharge summaries [3] |
| 2018 | Track 2: Adverse Drug Events and Medication Extraction | 505 discharge summaries |

[1] 2016, 2019 and 2022 dataset is not available for download on DBMI Data Portal.
[2] Only part of the original 2010 data is available for research beyond the original challenge.
[3] The dataset used in Track 1 of the 2018 n2c2 shared task consisted of longitudinal records from 288 patients, drawn from the 2014 i2b2/UTHealth shared task corpus.

---

recall through an iterative keyword expansion process. In each iteration, we rank content words in retrieved sentences by frequency and manually add high-frequency mobility-relevant keywords not included in the previous iteration. For example, "gait" and "adls" (activities of daily living) are absent in the ICF descriptions but were added to the keyword set through our iterative procedure. After five manual iterations, we obtain a set of 200 keywords, including inflections. Using this final keyword set on the 271,827 unique sentences above, we narrow down to 22,894 mobility-relevant sentences as our unlabeled data pool for subsequent active learning.

## B. Manual Annotation

The conventional active learning procedure starts with a small seed set of data, which is annotated to build initial mobility recognition models. In our study, we reuse parts of the annotation guidelines [10], taking portions related to the five entity types: Mobility, Action, Assistance, Quantification, and Score Definition. A domain expert then manually selects and annotates 120 sentences, ensuring that each one contains at least one Mobility entity.

In each iteration, we start by employing latest BERT model, trained on updated data from previous iterations, to pre-tag the present batch of unlabeled sentences selected through active learning. This pre-tagging step reduces manual annotation time, since annotators only correct existing tags rather than starting from scratch. To ensure accuracy, two human annotators follow a two-phase process. The first phase, called Blind Annotation, involves each annotator referring

to annotation guidelines to correct the machine pre-tagging errors. In the second phase, called Gold Standard Annotation, the annotators collaboratively resolve any discrepancies and achieve a consistent set of corrections. These additional gold standard labels obtained through the annotation process are then used to retrain the mobility NER model.

We implement each iteration within the time frame of one week. During the week, two medical students annotate a new batch of 125 sentences, with 100 added to the training set, and 25 to the validation set. The weekends will be allocated to training new mobility recognition models and re-scoring all remaining sentences in the unlabeled pool. Newly selected sentences from the unlabeled pool will be ready for the next week. All annotation activities are completed on the Inception platform [33].

*C. Active Learning*

*1) Methodology:* At each iteration, we apply active learning to select the most informative sentences for human annotation. We use a straight-forward pool-based query-by-committee sampling strategy [34]. A group of models, known as a "committee", evaluates the unlabeled pool and selects sentences on which they have the highest disagreement. Let $x = [x_1, ..., x_T]$ represents a sequence of length $T$ with a corresponding label sequence $y = [y_1, ..., y_T]$. NER models are trained to assign tags to each token in the input sequence x, indicating whether the token belongs to a particular type of mobility entities. We use vote entropy [34] as the base informativeness score:

$$\phi^{VE}(x) = -\frac{1}{T} \sum_{t=1}^{T} \sum_{m \in M} \frac{V(y_t, m)}{C} \log \frac{V(y_t, m)}{C}$$

where $C$ is the number of committee models, $M$ is a list that contains all possible label tags, and $V(y_t, m)$ is the number of "votes" or the level of agreement between committee members on assigning the tag $m$ to the token $t$.

To further improve the representativeness of selected sentences, we implement the information density metric proposed by Settles et al [18]. They defined the density score of a sentence as its average similarity to all other sentences in the unlabeled pool. The information density score is calculated as a product of the base informativeness score and the density score controlled by a parameter $\beta$:

$$\phi^{ID}(x) = \phi^{VE}(x) \times (\frac{1}{|\mathcal{L}|} \sum_{l=1}^{|\mathcal{L}|} sim(x, x^{(l)}))^{\beta}$$

To fit the limitation of our human annotator resource, we select top 125 sentences with highest information density scores for human labeling at the next active learning iteration. We refer readers to Settles[4] for a more comprehensive introduction of active learning.

---

[4] https://burrsettles.com/pub/settles.activelearning.pdf

*2) Named Entity Recognition Modeling:* We formulate the task of identifying mobility entities as a Nested NER problem since Action, Assistance and Quantification entities are encapsulated within the span of the Mobility entity. Thieu et al [10] proposed to use joined entity approach [35] to deal with nested entities, creating more complex tags by concatenating BIO (Beginning, Inside, and Outside) tags at all levels of nesting. While their approach has demonstrated good performance, it may not be suitable for our low-resource dataset. Creating more complex tags, e.g B-Action_I-Mobility, leads to sparsity in less frequent tags, making it challenging for NER models to learn and correctly identify these rare tags during training and inference. Therefore, we keep our active learning pipeline simple by training a separate model for each entity type using the BIO format where $M$ = [O, B-entity, I-entity].

*3) Model Choices:* Our active learning process, involving a human-in-the-loop approach, spans several months with weekly iterations. To ensure consistency, we select the committee models at the beginning and do not change them throughout the process. We chose BERT [19] and CRF [20] models for their ease of implementation and fast training and inference times, which allow sufficient time to re-train the models and score all candidates in the remaining unlabeled pool during each iteration. Additionally, previous research [10] demonstrated that the BERT model achieves the highest recall but lowest precision, while the CRF model has the lowest recall but highest precision for identifying mobility entities. These complementary characteristics make BERT and CRF well-suited for our active learning pipeline.

*4) Disagreement Signal:* From a theoretical perspective, Action is the central component of mobility information. A relevant sentence should contain at least one Action or Mobility entity while unnecessarily containing any Assistance or Quantification entity. As such, it is theoretically appropriate to compute the disagreement score based on Action NER models. From an empirical perspective, Action entities are shorter and easier to identify, while Mobility entities tend to encompass entire clauses or sentences [7], making them more challenging for sequence labeling models in a low-resource setting. We further empirically disregard disagreement signal from Assistance and Quantification because of their trivial predictive accuracy in the initial dataset. Specifically, our initial NER dataset contains 27 Assistance, 33 Quantification and no Score Definition entities. The numbers became even smaller after splitting the dataset into train and validation sets, making it insufficient to train NER models. Initial evaluation shows zero F1 scores on BERT models trained for these two entity types (Figure 3). Based on both theoretical and empirical observations, we choose to only rely on Action NER models for computing the disagreement score in active learning.

*5) Information Density Score:* The density score requires pairwise similarity calculations between sentences in the unlabeled pool. We use Sentence Transformers [36] loaded with Bio+Discharge Summary BERT weights to encode each sentence into an embedding vector. These vectors are used to compute cosine similarity scores between sentences. Information

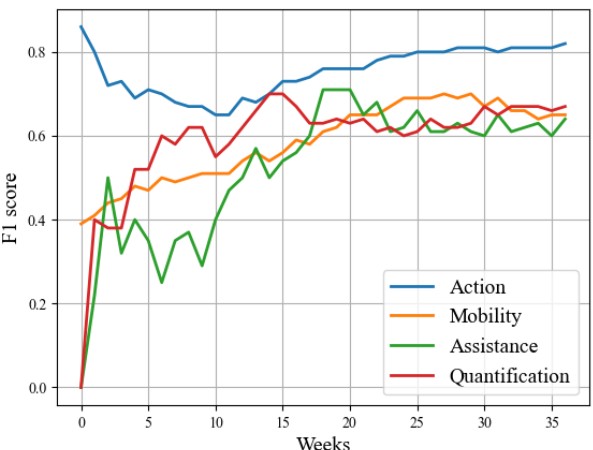

Fig. 3. Results of weekly BERT model for each entity type.

TABLE II
INTER-ANNOTATION AGREEMENT BETWEEN 2 ANNOTATORS (A AND B)
AND GOLD STANDARD DATASET. NOTE: E = EXACT MATCHING, P =
PARTIAL MATCHING

|  | Act | | Mob | | Ast | | Quant | |
| --- | --- | --- | --- | --- | --- | --- | --- | --- |
|  | E | P | E | P | E | P | E | P |
| A vs B | 0.8 | 0.92 | 0.73 | 0.93 | 0.7 | 0.91 | 0.65 | 0.89 |
| A vs Gold standard | 0.9 | 0.95 | 0.87 | 0.97 | 0.88 | 0.96 | 0.79 | 0.94 |
| B vs Gold standard | 0.87 | 0.95 | 0.78 | 0.96 | 0.73 | 0.92 | 0.74 | 0.94 |

TABLE III
NUMBER OF ENTITY MENTIONS IN OUR PUBLIC DATASET AND NIH
PRIVATE DATASET.

| Dataset | NIH [10] | Our |
| --- | --- | --- |
| Public | No | Yes |
| Source | 400 NIH PT notes | 4,265 n2c2 sentences |
| Action | 4527 | 5511 |
| Mobility | 4631 | 5328 |
| Assistant | 2517 | 306 |
| Quantification | 2303 | 639 |
| Score Definition | 303 | 0 |
| Total | 14281 | 11784 |

density metric, however, is computational demanding such that the number of required vector similarity calculations grows quadratically with the number of sentences in the unlabeled pool. For efficiency, we pre-compute density scores for all sentences offline only once and store these results for quick lookup during the active learning process. Finally, we set the controlled parameter $\beta$ to 1.

## IV. RESULTS

### A. Monitoring metrics

Figure 3 shows the performance of the BERT model for each entity type on the weekly validation set, which is regularly updated during the active learning process. Adding weekly curated data into the existing gold standard dataset results in fluctuation of the F1 score due to the change in the data distribution and the introduction of additional noise. It turns out only Action model maintains stable improvement throughout the active learning process. Performance fluctuates greater on entity types with a small number of instances such as Assistance and Quantification.

### B. Gold Standard Dataset

After repeating the active learning cycles for 9 months, we obtain a dataset comprised of 4,265 sentences that includes 11,784 entities (Table III). For quality assurance, we measure inter-annotator agreement (IAA) of entity mention spans using $F_1$ scores. Table II reports IAAs between two annotators and between each annotator versus gold standard adjudication using exact matching and partial matching. The average exact matching scores between the two annotators for all entity types are 0.72, indicating a moderate level of agreement in identifying the exact boundaries of the entities. Furthermore, the average gap of 19% between exact matching and partial matching across the two annotators is relatively large. It suggests that while identifying the presence of a mobility-relevant entity in a sentence is easy, accurately determining its span boundary is more challenging.

There are two main differences in the distribution of entities between our data set and the closest work, the NIH private dataset [10]. First, we do not detect any instance of the Score Definition entity. It appears that the Score Definition entity is unique to PT notes at the NIH and is therefore not observed in the n2c2 notes. As a result, all of our model training and evaluation exclude this entity from consideration. Second, we observe significantly smaller number of Assistance and Quantification entities in our dataset. This disparity exists because n2c2 notes are not focused on physical therapy, unlike PT notes, where mobility and functional information are more commonly documented. While this broadens the generalizability of our dataset for diverse applications, it also introduces challenges for NER model training, particularly in addressing class imbalance and low-resource entity types. Despite these challenges, our public dataset provides a first-ever, valuable resource for advancing NLP research in mobility and functional status domains.

## V. BENCHMARKING

We benchmark our public dataset using vanilla BERT [19] and advanced Nested NER methods [37], [38] to evaluate its utility for extracting mobility-related information from clinical text. To ensure a robust evaluation, we perform benchmarking using five-fold cross-validation on our finalized gold standard dataset. Specifically, we use StratifiedKFold function from scikit-learn library [39] to split our dataset into five balanced subsets, ensuring that each subset contains a similar number of instances for each entity type. For each fold, we use three subsets for training, one subset for evaluation and one subset for testing. The final $F_1$ score is the average of the five-fold scores.

TABLE IV
AVERAGE F1 SCORE FOR FIVE-FOLD CROSS-VALIDATION EXPERIMENTS.
NOTE: DS BERT=DISCHARGE SUMMARY BERT,
ALBERT=ALBERT$_{XXLARGE-V2}$

| Method | Pretrained weights/embeddings | Act | Mob | Ast | Quant |
|---|---|---|---|---|---|
| BERT [19] | BERT$_{base}$ | 81.13 | 63.16 | 51.76 | 61.96 |
| | BERT$_{large}$ | 81.88 | 65.52 | 53.21 | 63.60 |
| | DS BERT | 81.09 | 63.54 | 54.66 | 65.73 |
| | Gatortron$_{base}$ | 82.77 | 67.14 | 52.43 | 63.04 |
| Pyramid [37] | ALBERT + BERT$_{base}$ | 82.75 | 67.38 | 58.89 | 66.93 |
| | ALBERT + BERT$_{large}$ | 82.28 | 67.35 | 59.87 | 66.93 |
| | ALBERT + DS BERT | 82.73 | 68.48 | 57.29 | 67.02 |
| | ALBERT + Gatortron$_{base}$ | **83.30** | **68.54** | 58.17 | **67.17** |
| BINDER [38] | BERT$_{base}$ | 81.76 | 66.58 | 54.87 | 64.82 |
| | BERT$_{large}$ | 81.54 | 66.84 | 52.84 | 58.94 |
| | DS BERT | 81.72 | 66.16 | 56.22 | 65.04 |
| | Gatortron$_{base}$ | 83.08 | 67.68 | **60.02** | 63.04 |

To evaluate performance, we first train a separate model for each entity type using BIO format as mentioned in III-C2 and compare the performance using four pretrained language models: BERT$_{base}$, BERT$_{large}$ [19], Bio+Discharge Summary BERT [40] and Gatortron$_{base}$ [41]. Considering the nesting structure of the entity types, we further apply two state-of-the-art nested NER methods: Pyramid [37] and BINDER [38], which have demonstrated superiority on well-known datasets such as ACE04 [42], ACE05 [43], and NNE [44].

Table IV presents the F1 score for each entity type, averaged across five-fold cross validation. Training a single nested NER model for all entity types yields better performance than training individual models for each entity type. The best performing nested models achieve F1 scores of 0.83 for Action, 0.69 for Mobility, 0.60 for Assistance, and 0.67 for Quantification. The results also show that transferring knowledge from a language model pre-trained in the clinical domain is beneficial. Specifically, models that yield the highest F1 scores for each entity type are fine-tuned or utilize embeddings derived from Gatortron$_{base}$ [41].

Model performance is proportional to the size of the training data for each entity type. For example, all models achieve high accuracy with Action entities while struggling with data sparsity of Assistance and Quantification entities. In addition, lengthy spans and token variability in Mobility entities are barriers to exact identification.

## VI. DISCUSSION

### A. Comparison to Related Works

Disregarding the difference in data distribution and model architecture, entity recognition performance on our dataset is slightly lower than on the NIH private dataset [10], with a 2-4% performance gap for Action and Mobility entities, and over 8% for Assistance and Quantification entities. These gaps can be explained by three main reasons. First, our dataset is more challenging due to the diversity of language use in the n2c2 research dataset, which includes clinical notes from 15 different hospitals and healthcare institutes. In contrast, the NIH private dataset only contains physical therapy notes collected at the NIH. Second, the NIH private dataset is annotated by senior experts, whereas our dataset is annotated by medical students, including one master's and one PhD student. The discrepancy in experience and expertise might contribute to a lower IAA in our annotation. Lastly, our dataset is largely imbalanced with low-resource Assistance and Quantification entities, leading to challenges in training and evaluating NER models for these entities.

We also scan our dataset for overlap of mobility terms compared to a dictionary recently published by Zirikly et al [28]. Our dataset includes 3,525 sentences that each contains at least one Mobility entity. Scanning these sentences against 2,413 mobility terms provided in the NIH dictionary, we found 907 sentences that do not contain any NIH mobility term. For example, a phrase "able to salute and brush teeth with either hand" is annotated in our dataset with ICF codes *d440 - Fine hand use* and *d445 - Hand and arm use*. However, the NIH dictionary [28] only considers "brush teeth" in a self-care context with ICF code *d520 - Caring for body parts*, thus missing its mobility context. Another example is that a keyword search using the NIH mobility terms will miss the sentence "She was able to go two flights without extreme difficulty" because the generic verb "go" is not included.

### B. Future Direction

We plan to apply in-context learning with pretrained large language models (LLMs) [45], [46] to address scenarios where our dataset contains insufficient instances of Assistant or Quantification entities for effective supervised fine-tuning. Although powerful, LLMs (including ChatGPT) face challenges in determining the boundary characters of entities and struggle to adhere to the instruction not to rephrase the extracted entity text [47]. These limitations highlight areas where further improvement is needed to enhance in-context learning for low-resource entity types.

## VII. CONCLUSION

In this study, we annotate the first publicly available dataset to train and evaluate NER models that extract ICF's Mobility-related information from clinical notes. We also benchmark popular and cutting-edge NER methods on the dataset. We hope that releasing the dataset to the research community will accelerate the development of methodologies to identify the complex spectrum of information about whole-person functioning in EHRs.

## ACKNOWLEDGEMENTS

We would like to thank Suhao Chen for pre-processing n2c2 research datasets and Thanh Duong for installing the Inception annotation platform.

## DATA AVAILABILITY

The n2c2 research datasets are available at (https://portal.dbmi.hms.harvard.edu/projects/n2c2-nlp/) to researchers who signed NLP Research Purpose and Data Use Agreement form.

Our Mobility dataset can be downloaded from our research group's website at https://lailab.info/research. To access the dataset, select Whole-person function and click View Data Source then follow the provided instructions to complete the download process. Our code is available at: https://github.com/dzunglt24/Active_Learning_Mobility

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
