# OpenReview forum: "Leveraging deep active learning to annotate the first public dataset for identification of mobility functioning information in clinical text"
_IEEE.org/EMBS/BHI/2025/Conference — BHI 2025_

### Official Review · Reviewer_tGY3 · 2025-06-26
**A valuable public dataset contribution with strong methodology, but minor limitations in clinical grounding and entity balance.**

**Confidence:** 4
**Clarity Of Writing:** great
**Clinical Significance:** good
**Methodological Novelty:** great
**Overall Rating:** 7

**Experiments And Results:**

great

**Questions For The Authors:**

1. Downstream Use Case: Could you elaborate on how this mobility information will be utilized in real-world clinical systems? Would EHR vendors or clinicians have a practical use for it today?
    - A detailed response showing integration pathways would positively influence the assessment of clinical significance.

2. Score Definition Entity: Given that Score Definition was completely missing from the n2c2 corpus, how would you propose expanding the dataset in the future to include this (e.g., by combining with MIMIC-IV)?
    - Exploring expansion strategies could improve long-term dataset utility.

3. Entity Overlap and Confusion: Have you observed common errors or confusion between overlapping entities (e.g., Action vs. Mobility)? Would joint modeling help?
    - If resolved effectively, this could support a higher methodological novelty score.

4. Model and Annotator Efficiency: How long did it take per batch of 125 sentences for annotation? Could you share insights into annotator fatigue or guidelines drift over time?
    - A response would add credibility to the reproducibility and annotation quality claims.

**Strengths:**

1. Timely and Meaningful Contribution: This is the first public dataset dedicated to mobility-related ICF entity extraction, addressing a major gap in clinical NLP.
2. Methodological Rigor: The use of query-by-committee active learning, paired with density-based sampling and strategic annotation, is well-motivated and effective.
3. Robust Benchmarking: The paper evaluates multiple NER models using five-fold cross-validation and provides clear comparisons.
4. Transparency: Inter-annotator agreement (IAA) metrics are reported and discussed, and the annotation process is well-documented.
5. Reproducibility: Code and data (with placeholders) are planned for public release, which supports the BHI community's goals.

**Summary Of The Paper:**

This paper presents the first publicly available dataset for the identification of mobility functioning information in clinical notes, based on the Mobility domain of the International Classification of Functioning, Disability and Health (ICF). Using the n2c2 dataset as a source, the authors employed a deep active learning approach with query-by-committee sampling (BERT and CRF models) weighted by density representativeness to select informative sentences for manual annotation. Over nine months, 4,265 sentences were annotated, resulting in a dataset with 11,784 entities. The authors benchmarked various Named Entity Recognition (NER) models (BERT, Pyramid, BINDER) on the dataset and demonstrated promising F1 scores, especially for Action and Mobility entities. The dataset and code are to be released publicly to facilitate further research.

**Weaknesses:**

1. Limited Clinical Insight and Impact Discussion: While the technical novelty is strong, the paper could better articulate the downstream clinical significance of extracting these mobility entities. For example, how would this facilitate better decision support, patient outcome prediction, or integration into clinical workflows?

2. Class Imbalance: Assistance and Quantification entities are underrepresented in the dataset, and while acknowledged, this substantially limits model utility for these subtypes.

3. Nested Entity Modeling Not Fully Explored: Although nested NER is an important aspect, the paper defaults to training separate models per entity, possibly missing synergy between them.

4. Generalizability: While the n2c2 data improves institutional diversity, it is not shown how well the trained models would generalize to newer datasets like MIMIC-IV or clinical notes outside of discharge summaries.

5. Clarity in Figures: Figures (especially Figure 3) are not well-annotated in the text; their descriptions are brief and can be more interpretable.

---

### Official Review · Reviewer_Bkhs · 2025-07-03
**Deep active learning review**

**Confidence:** 3
**Clarity Of Writing:** great
**Clinical Significance:** good
**Methodological Novelty:** fair
**Overall Rating:** 6
**Final Rating:** 7

**Experiments And Results:**

great

**Questions For The Authors:**

-Could you explain why ICD is used more in practice compared to ICF? What advantages does it have? If it is used more in practice, is the fact that WHO backs ICF a good enough justification for using it instead of ICD?

-“Since function is not well perceived in medical coding…” – Why is this the case?

-Were class weighting schemes used when training models using your curated dataset?

-Your downstream models have poor scores for predicting Assistance. Why do you think that is?

**Strengths:**

-The introduction provides ample justification for the study, clearly outlining the clinical problem as well as the lack of public datasets for this type of problem.

-Well designed process to reduce complexity of the dataset for annotation. I like how specific the authors are in describing their process, I feel confident that someone could replicate their study given the detail provided here.

-Results are well presented with many comparisons. I like the comprehensive evaluation of the dataset as well as the downstream models chosen to benchmark on it. The explanation of the results is also grounded well. The example given of how prior datasets fail to address the functional implications of specific notes is enlightening.

**Summary Of The Paper:**

The paper addresses the lack of structured functional‐mobility data in clinical text by developing an automated pipeline to identify and label mobility mentions in discharge summaries. After splitting notes into sentences and filtering them via a keyword index, the authors apply a pool‐based deep active‐learning loop to iteratively select and annotate the most informative examples. The final output is a publicly released dataset of 4,265 sentences and trained models that achieve agreement levels comparable to expert annotators.

**Weaknesses:**

-While the dataset may be unique, the method used in this study is not. Deep active learning is relatively common, limiting the novelty of this specific study.

-I am not convinced by the “computational feasibility” argument for only using n2c2. Why not compile all the data if its available? Your proposed method is meant to limit the annotation burden, so it would significantly strengthen the study to show its effectiveness on a larger dataset. Can you explain exactly how much more time would be required to annotate this additional data?

-The class imbalance of the filtered dataset may hamper its usefulness in further tasks. What impact could this have on future studies? Similarly, if the matching scores between your annotators was so different, can the final labels be trusted for downstream training?

---

### Official Review · Reviewer_v19j · 2025-07-16
**Leveraging deep active learning to annotate the first public dataset for identification of mobility functioning information in clinical text**

**Confidence:** 3
**Clarity Of Writing:** good
**Clinical Significance:** good
**Methodological Novelty:** fair
**Overall Rating:** 6

**Experiments And Results:**

fair

**Questions For The Authors:**

1. Could you clarify the span‑boundary rules in your annotation guidelines and provide a few illustrative examples?
2. How many sentences would random sampling have required to reach the same dev‑set F1, and what would that translate to in annotation hours?
3. Have you evaluated the trained models on an external corpus such as MIMIC‑IV, and if so, how did performance compare?

**Strengths:**

1. The paper provides the first publicly available dataset focused on ICF mobility, addressing a clear gap in clinical NLP.
2. Annotation workflow uses weekly machine‑assisted rounds and double‑blind adjudication, achieving solid agreement scores.
3. Active‑learning procedure is described clearly, making it feasible for replication.
4. Experiments compare flat and nested NER models with both general and clinical embeddings, giving useful baselines.

**Summary Of The Paper:**

The authors compile and release a corpus aimed at extracting ICF mobility information from clinical notes. Starting with 22 894 mobility‑relevant sentences, they run a nine‑month active‑learning loop that combines BERT and CRF committees. Two annotators label 4 265 sentences for four nested entity types and achieve a partial inter‑annotator agreement of 0.91. Baseline models, including Pyramid and BINDER nested NER with GatorTron embeddings, reach F1 scores of 0.83 for Action and 0.69 for Mobility.

**Weaknesses:**

1. Assistance and quantification entities are underrepresented, limiting model performance for those classes.
2. Keyword‑based sentence selection may miss less obvious mobility expressions, introducing selection bias.
3. The study does not measure annotation savings against simpler sampling methods, leaving the efficiency of the active‑learning strategy unquantified.